# Tasting to preserve: An educational activity to promote children's positive attitudes towards intraspecific diversity conservation

**Patrícia Pessoa**[1,2]*, **Sara Aboim**[3], **Lisa Afonso**[4], **J. Bernardino Lopes**[1,2], **Xana Sá-Pinto**[1]

**1** Research Centre in the Didactics and Technology in the Education of Trainers of University of Aveiro (CIDTFF.UA), Aveiro, Portugal, **2** University of Trás-os-Montes e Alto Douro (UTAD), Vila Real, Portugal, **3** P. Porto: School of Education and Centre for Research and Innovation in Education (inED), Porto, Portugal, **4** Center for Psychology, Faculty of Psychology and Educational Sciences of the University of Porto, Porto, Portugal

* afppatricia@gmail.com

**Data Availability Statement:** All data files are available at https://doi.org/10.48527/QQYR4I.

**Funding:** Patrícia Pessoa and Xana Sá-Pinto are funded by Portuguese national funds through FCT

## Abstract

On the edge of causing the sixth big mass extinction event, the development of positive attitudes towards the conservation of intraspecific diversity from early ages is essential to overcome the biodiversity crisis we currently face. However, there is no information available on elementary school students' attitudes toward intraspecific diversity conservation nor is there a framework available to perform such analysis. For this study we designed, implemented, and evaluated an educational activity planned for third graders (8 to 13 years old) to explore the intraspecific diversity of vegetables and promote healthy eating habits. Additionally, a framework was developed to evaluate students' attitudes towards intraspecific diversity conservation and applied to semi-structured interviews conducted with students before and after engaging in the educational activity. In this paper we present a reliable framework, developed aligned with the ABC model of attitudes, based on literature, and adapted to elementary school students' responses, to evaluate students' attitudes toward intraspecific diversity. Our results show that, before the educational activity, most students choose a non-biodiverse option, justifying this choice with the affective component of attitudes: mostly emotional factors but also aesthetic and social/cultural factors. After the educational activity, we observed a significant increase in the frequency of students that choose the biodiverse option and that justified it with the cognitive component of attitudes: mainly with biology and health knowledge factors, but also with economic and ethical knowledge factors. Our findings support the positive impact of educational activities that explore vegetable varieties on students' attitudes toward intraspecific diversity conservation. This activity may also be used to foster education for sustainability and address socioscientific issues aligned with diverse sustainable development goals.

– Fundação para a Ciência e a Tecnologia, I.P. (https://www.fct.pt/), within the scope of the PhD grant 2020.05634.BD, and of the framework contract foreseen in the numbers 4, 5 and 6 of the article 23, of the Decree Law 57/2016, of August 29, changed by Law 57/2017, of July 19, respectively. This work is financially supported by National Funds through FCT – Fundação para a Ciência e a Tecnologia, I.P. (https://www.fct.pt/), under the project UIDB/00194/2020. The funders had no role in study design, data collection and analysis, decision to publish, or preparation of the manuscript.

**Competing interests:** The authors have declared that no competing interests exist.

## Introduction

Life on earth went through five big mass extinction events—short geological periods during which three quarters of existing species went extinct—and studies on extinction rates show that we are at the edge of causing the sixth one [1]. This high extinction rate can be observed across diverse types of ecosystems [2–4] and taxonomic groups [5–7]. Biodiversity crisis has multiple causes that are linked and reinforce each other in complex systems that act at both local and global levels [5]. Mostly driven by human overpopulation and overconsumption, the major causes of the extinction include habitat reduction, fragmentation and reconversion, climate change, overexploitation, invasive species, diseases, toxification and emerging diseases [5]. Researchers have been warning us about the impacts that our actions may have in current and future generations [8]. This biodiversity crisis affects ecosystems' equilibrium and may compromise our own survival, negatively impacting the services provided by ecosystems, such as the supply of food and other goods or the regulation of ecosystem functioning, leading to feedback loops that further increase the rate of environmental change [9]. This biodiversity emergency is now globally recognized as one of the major socioscientific issues (SSI). In fact, several of the 17 sustainable development goals (SDGs) established by the United Nations [10], such as 'Zero hungry', 'Good health and well-being', 'Responsible consumption and production', 'Climate action', 'Life below water' or 'Life on land' (SDGs 2, 3, 12, 13, 14, and 15 respectively), are strictly related to or affected by the biodiversity crisis.

Overcoming the biodiversity crisis and achieving the 17 SDGs, requires public understanding of biodiversity and the factors that threaten it as well as the development of positive attitudes towards its conservation. According to the International Convention on Biological Diversity [11 p16], biodiversity is "the variability among living organisms from all sources including, inter alia, terrestrial, marine and other aquatic ecosystems and the ecological complexes of which they are part; this includes diversity within species, between species and of ecosystems". Studies show that the importance of biodiversity is recognized and understood by both adults and children, but people tend to focus on the importance of the number of species within an ecosystem, with less attention being paid to the ecosystems' diversity or genetic diversity at the intraspecific level [12–14]. But, although less publicly recognized, intraspecific diversity is critical in decreasing the probability of extinction by increasing its ability to adapt to environmental changes [15–17]. On the contrary, reduced genetic diversity in a population or species decreases its ability to adapt to short (such as a new disease or a severe drought for example) and long term habitat changes (such as climate change for example) increasing the probability of its extinction [18, 19]. In addition, intraspecific diversity plays a crucial role in the regulation of ecological processes [20], and is essential to ensure food security, and medicine development, to preserve cultural values and diversity and as a source of inspiration [16]. In terms of food security, results suggest that intraspecific diversity in agricultural plants may strongly reduce the losses expected in agricultural regions due to global warming [21], contribute to control of agricultural pest species affecting crops [22, 23] and may enhance the dietary intake of diverse nutrients and influence consumers preferences [24, 25]. It is thus fundamental that people develop positive attitudes towards conserving intraspecific diversity, and schools represent an ideal setting to develop this awareness [26].

To develop positive attitudes towards intraspecific diversity conservation we should consider that attitudes are a response to a prior stimulus [27]. According to Breckler (1984), attitudes are defined as a tripartite model, the ABC model, that includes three components, affect (A), behavior (B) and cognition (C). However, with regard to the factors, there is no consensus on which ones lead individuals to act or not to act towards biodiversity conservation or, even more broadly, towards environmental protection [28–30]. Several factors have been shown to

influence individuals' actions towards biodiversity conservation, such as demographic, institutional, economic, social and cultural features, knowledge, attitudes, motivation, awareness, values, emotion, responsibility, priorities, childhood experiences, activity choices, personality, perceived behavioral control, behavioral intention, among others [28–30]. Despite the lack of consensus, studies have shown that knowledge predicts behavior, although it is considered a necessary but not sufficient condition for decision-making [31, 32]. This is aligned with the elaboration likelihood model (ELM) [33] developed to explain attitude changes. This model predicts that the likelihood and strength of attitudes' changes increase with the degree of time and effort spent by a person on the interpretation of and cognitive building around information important to evaluate the attitudes' object [33]. When stronger elaborative processes are promoted and people engage with strong arguments, ELM predicts attitudes' changes that are persistent over time, resistant to counter persuasion and predictive of behavior [33]. This highlights a potential role for science education to influence students' attitudes, by allowing students to engage with and elaborate around strong arguments and contributing, among other things, to emotional engagement and knowledge acquisition, which is expected to directly impact the affective and cognitive component of attitude respectively.

Some studies have shown that educational activities can have a positive impact on students' attitudes, such as towards human parasites [34], birds [35], reptiles [36] and conservation of extensive grasslands [37]. Furthermore, a survey was administered to elementary school students showed that factors such as gender, species knowledge, preferred leisure activities, and family members' involvement in nature protection organizations are a significant predictor of children's attitudes toward insects [38]. Concerning biodiversity conservation, a study by Rosalino et al. [39], showed that elementary school students' interests and values related to health and economy may override positive attitudes towards biodiversity conservation. In addition, the same research concluded that students from urban areas are less likely to have a pro-conservation attitude and that they give conservation priority to the species that are most frequently mentioned in online news, particularly mammals and plants. However, to the best of the authors' knowledge, no study examined elementary school students' attitudes towards intraspecific diversity conservation nor is there a framework available to perform such an analysis, despite recent studies showing that younger children have a scientifically acceptable understanding of intraspecific diversity [17, 40]. Filling this gap can be fundamental to inform the development of educational activities, school curriculum documents, and teacher training.

Food species have observable intraspecific biodiversity attributes (such as taste, shape, and texture) which children can easily explore in hands-on activities [41, 42]. Healthy eating education is addressed in school curricula in several countries [43–45] and our research team developed an educational program that explored the intraspecific biodiversity of tomatoes and that fostered children's acceptance of that vegetable [46]. Given this, we hypothesize that exploring intraspecific diversity in food plant species can also promote elementary school students' positive attitudes towards its conservation. However, and although some studies analyzed the impact of exploring intraspecific diversity to promote healthy eating [47], no studies analyzed the potential of exploring food to promote positive attitudes towards intraspecific diversity conservation.

With the present study we aim to answer two research questions: 1) what are the elementary school students' attitudes towards intraspecific diversity conservation? 2) how educational activities that explore intraspecific diversity in a food specie affect these attitudes. To answer these questions, in this study, we i) developed an analysis framework to identify the factors that influence elementary school students' attitudes toward intraspecific diversity conservation, ii) identified the factors that influence elementary school students' attitudes toward intraspecific diversity conservation, and iii) studied the impact of an inquiry based learning

educational activity designed to increase students' knowledge related to the importance of intraspecific diversity in their attitudes towards intraspecific diversity conservation.

## Materials and methods

To achieve our goals, we implemented a randomized controlled trial. We invited third-grade children (8-13-year-old) from one public elementary school in the northern region of Portugal, chosen by convenience [48], to attend the educational activity and answer the short interview described below (Girls: 41.6%; Age: M = 8.88; SD = 0.65). Parents of eligible children (62 from 3 classrooms) were asked to allow their children to participate. From the 65 children invited, we collected data from 58 children due to exclusions caused by withdrawal (n = 2), school transfer (n = 1) and missing post-test (n = 1). The children were randomly distributed (1:1) within the classrooms, using a random number generator, to form a target group (n = 33) and a control group (n = 25). The parents of all the students enrolled in this study gave their written informed consent. This study was conducted according to the guidelines laid down in the Declaration of Helsinki and all the procedures were approved by the Ethics Committee of the University of Aveiro (Process code 10/2018) and by the school board.

### Educational activity

The educational activity was designed for third-grade students following an inquiry based learning and experiential approach [49]. It was developed to address several learning objectives (view [46] for more detailed information about this activity), the most important of which were to promote the valorization of intraspecific diversity of vegetables and healthy eating habits. In accordance with Portuguese elementary school curricula and European recommendations for these school years, this educational activity, addressed learning objectives related to environmental literacy, scientific literacy and health literacy, such as: i) discussions about biodiversity loss and conservation and its impacts on our environment and food security [43], ii) discussions about the importance of a healthy diet and the importance of the correct chewing for proper functioning of the digestive system [43], and iii) engagement in scientific practices, by planning and implementing experiments [43, 50, 51].

The educational activity comprised three sessions, lasting 60–90 min each (total intervention duration: 240 min), with 10 to 20 children and were implemented in the schools from April to June 2018. In the first session, students were introduced to three varieties of tomato (beef, plum and cherry tomato). They were invited to explore the visual differences, to taste slices of each variety and to classify their sweetness and acidity, in a 5-point Likert scale (from not sweet/acid to very sweet/acid, respectively). Based on students' classifications six bar graphs (one per variety for acid and sweet flavors) were constructed and their meanings discussed with the class.

In the second session, students analyzed the bar graphs and postulated hypotheses to explain two observations that were posed to them to be answered through an inquiry-based learning approach [49]: a) why different varieties of tomato were classified as having different levels of sweetness and acidity; and b) why different individuals classified differently the sweetness and acidity of the same variety of tomatoes. In groups of 3 to 5, students were invited to plan an experiment to test their chosen hypothesis (examples of hypotheses that were tested can be found in [46]) using a worksheet adapted from an official Portuguese educational set [52]. The worksheet asked them to describe the hypothesis rephrased as a question, what would need to be kept constant to test the hypothesis, what would vary, what was going to be measured, how were they going to register data, what would be the experimental procedure, what was needed to perform such experimental procedure, what were the predicted and

observed results, and what conclusions could be taken. During this session, the researchers supported each group in the identification of problems in their experimental design and promoted discussion to find solutions to overcome these. The devices used to measure sugar (refractometer) and pH (pH sensor) were shown to the students whenever necessary [52].

In the third session, each group implemented their experimental plan, collected data, analyzed their results, and discussed how these supported or not the initial hypothesis (examples of experimental procedures used to test each hypothesis can be found in [46]). After this was done in each group, a discussion was held with the whole class in which other vegetable species with diversity were presented, such as *Brassica oleracea* (kales, cabbage, broccoli), *Malus domestica* (apples), *Daucus carota* (carrots). The students' preferences for the different varieties and the consequences of the absence of the different varieties on the cuisine and whether the students could eat that vegetable were discussed.

While planning and implementing this activity we took into consideration the ELM model of attitudes' change [33]. These elements were aimed to increase the students' elaborative processes and the outcomes of these by: *i)* increasing students' motivation (by making the information personally relevant and needed by using vegetables that have several varieties familiar to students, that are often used in Portuguese cuisine, which are easy to find in supermarkets and which students were invited to taste and classify according to their own perceptions), *ii)* increasing their ability to process the information (by creating diverse opportunities to experience—in the first and third sessions—and discuss the intraspecific diversity and its value- in the three sessions); and by *iii)* providing them diverse and strong arguments to value intraspecific diversity (by addressing individual emotional preferences, aesthetic and social/cultural values, and biology/health, economic, and ethical knowledge). Namely, during the activity, individual emotional preferences were explored by asking students for their own preferences regarding the diversity of various vegetables. Aesthetic values were explored through the colors of the presented vegetables. Social/cultural values were explored through the importance of intraspecific diversity for Portuguese gastronomy, but also for the traditional gastronomy of other countries accounting for the students' families. Biology and health knowledge covered topics such as: 1) different varieties of vegetal species have different properties [41, 42]—*i*) different varieties have distinct tastes; *ii*) different varieties may have distinct features that make them more suitable for distinct dishes; *iii*) different varieties have different nutritional properties and make our diet more diverse; 2) different varieties may grow and produce differently in distinct environments [42]; 3) the food's degree of ripeness alters its flavor [53]; 4) different people have distinct abilities to taste some flavors and have distinct preferences [54]; 5) it is healthy to eat distinct varieties of a vegetable and/or it is not healthy to always eat the same variety of a vegetable [55]; 6) our tastes change over time, so we should try different varieties of vegetables [56]; 7) the way we chew food influences the way we taste it [57]; 8) the fact that an individual is sick can change the way they taste food [58]. Economic knowledge was explored by reflecting on the impact of variety availability on the ability/need to buy and sell vegetables. And the ethical knowledge was explored through different people's preferences for different varieties and their right to assess these.

## The interview and its implementation with students

All students were interviewed twice. The students in the target group were interviewed before and after the educational activity. The students in the control group were interviewed in the same period as the students of the target group but only performed the educational activity after the post-test. A semi-structured interview [48] was conducted by one researcher (XSP) who asked students individually to i) pretend to be farmers and select seeds to sow in their

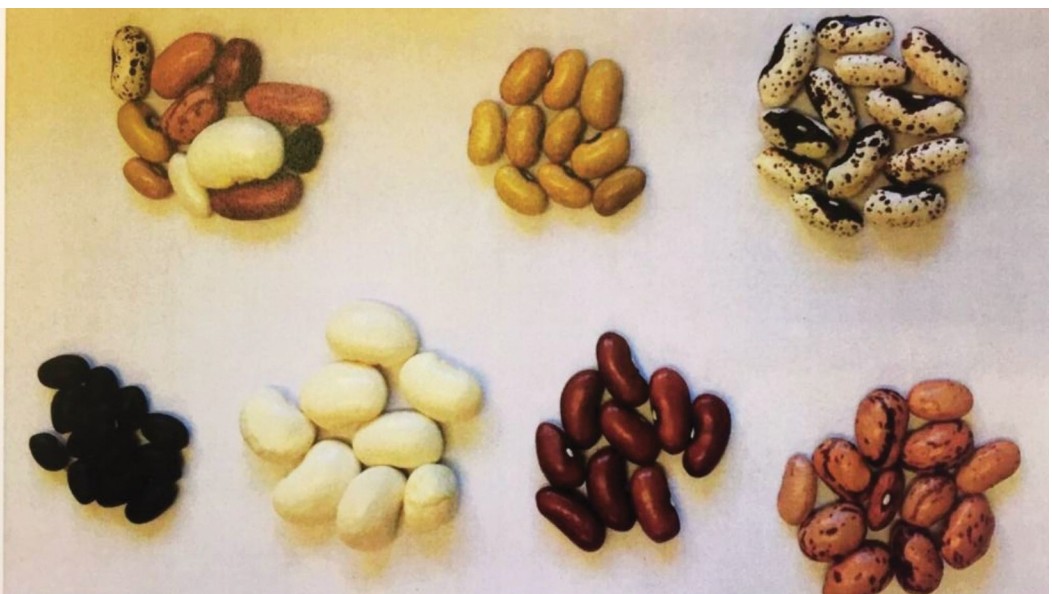

**Fig 1. Example of the bean varieties placed in the seven bag options.**

imaginary farm from 7 options of bean seed bags each containing a single variety and a biodiverse bag containing seeds from the seven varieties (see Fig 1) and ii) justify their choice. After the first verbalization of the choice and justification, the researcher asked follow-up questions and probing questions [48] to encourage the students to elaborate more on their justification and, whenever possible, to explain their choice in more depth. These questions were always supplemented by interpretation questions [48] to ensure that the researcher understood the student's justification. Interviews ended with a question to confirm that there were no reasons not previously mentioned to justify the student's choice. All responses were audio recorded, transcribed, and anonymized after the collection of the post-test data (from June to August 2018), after which all recordings and files containing personal data were destroyed.

### Design and application of the analysis framework

To assess the students' attitudes towards intraspecific diversity and the factors that lead those, we developed and applied a framework to perform a content analysis of the answers to the interview. To develop the analysis framework we followed the ABC model of attitudes developed by Breckler (1984), that defined attitudes in three components, affect (A), behavior (B) and cognition (C) [27]. According to this model, the affective component (A) is defined as being an emotional response, an instinctive reaction, or sympathetic nerve activity, which in turn can be measured by monitoring physiological responses, such as heart rate, or by an individual's verbal report. The behavioral component (B) is defined by overt actions, behavioral intentions, and verbal statements regarding the behavior. The cognitive component (C) includes beliefs, knowledge structures, perceptual responses, and thoughts. To assess each of these components we divided the students' answers into two segments, choice, and justification (see Fig 2). In the choice segment we assess the student's behavioral intention through the verbalization of which bag of seeds they would buy to sow on their farm. In the justification segment we assess the affective and cognitive components by analyzing the factors stated by the students that influenced them to make a certain bag choice. To assess the factors that lead to the affective and cognitive components of elementary school students' attitudes we created

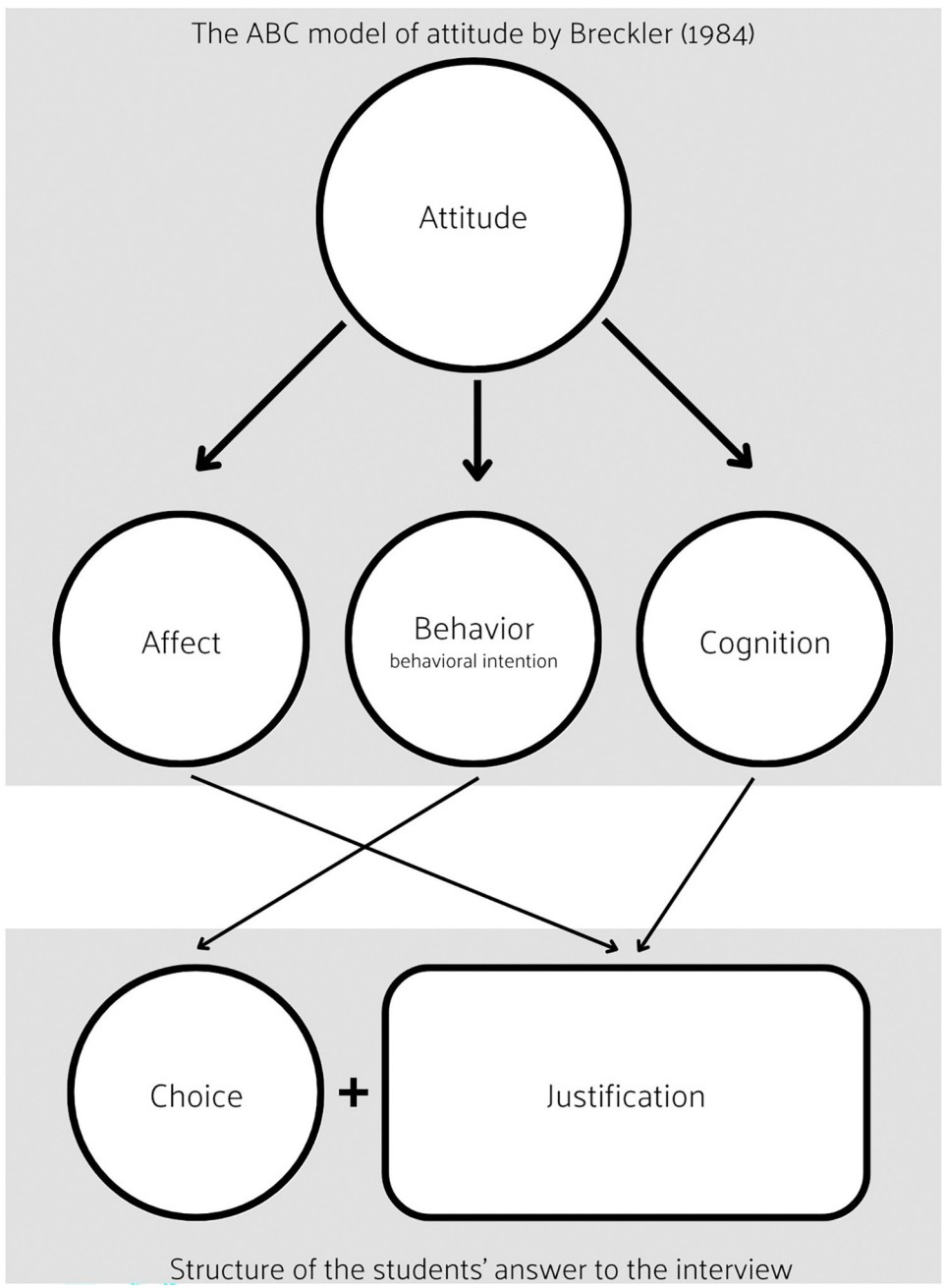

**Fig 2. Schematic of the links established between the ABC model of attitude and the structure of students' answers to the interview.**

categories based on previous studies on the values, meanings, and representations of biodiversity [59, 60], on the factors that lead individuals to act or not act pro-environmentally [29, 30], and on the floating reading of the students' answers accessed through the transcripts previously made (see Table 1). The first framework was developed in October 2018 and further improved through cycles of refinement and clarification of the categories to ensure its reliability and validity until June 2022.

All students' justifications were subjected to a content analysis [48] to evaluate the presence or absence of the categories of analysis in July 2022. Since attitudes are the result of the interaction of three components, each student's answer can be categorized within more than one category. The unit of analysis was the "meaning unit", which is defined as "the constellation of words or statements that relate to the same central meaning" [61 p106]. In our case, a meaning unit can consist of a sentence or sentence segment that expresses an idea and aligns with a specific category of the framework of analysis. To ensure the reliability of the framework, two researchers (PP and SA) analyzed 20% of the interview answers using the final framework presented here in October 2022. Interrater reliability was estimated as the percentage of the initial agreement between raters [62]. Since interrater reliability was > 87%, the reliability of this procedure was considered acceptable [63]. The coding of all students' answers was performed by one trained researcher (PP). The answers from the post-test from the students in the control group were used to check for the impact of students' double exposure to the test and to evaluate the internal process validity [64]. The pre- and post-tests of the control group did not significantly differ ($p > 0.05$ for all the analyses, see S1 Table for detailed results), thus confirming the internal validity of the process [64].

## Statistical analysis

To test for significant differences between pre- and post-tests results for each of the categories of analysis in control and target groups we performed McNemar tests. To study possible associations between affective and cognitive components with the students' behavioral intention in pre- and post-tests we used Chi-square tests. All statistical analyses were performed with SPSS v26.

## Results

### Framework of analysis

Our final framework of analysis is organized according to the three components of the ABC model of attitudes [27]. The affective component was subdivided into three categories (emotional, aesthetic, and social/cultural), the behavioral component was defined by the behavioral intention verbally indicated by the students when choosing the seed bag, and the cognitive component was subdivided into three categories of knowledge (biology and health knowledge, economic knowledge, and ethical knowledge). A description of the categories and examples of students' answers assigned to each category can be found in Table 1. The 'biology and health knowledge' category was defined by the learning goals addressed in the educational activity, and as such, this category may change in other implementation contexts. The learning objectives addressed in our educational activity and the full description of the category 'biology and health knowledge' applied in this study can be found in S3 Table.

### Students' attitudes towards intraspecific diversity conservation

In the pre-test, when we asked students to verbally indicate their behavioral intention, or in other words, which bag of seeds they would choose to sow on their farm, 9.09% of the students in the target group (TG) and 24% of the students in the control group (CG) chose the biodiverse bag. The analysis of the students' justifications in both groups revealed that this behavioral intention—the choice of the biodiverse bag—was followed by justifications that mentioned the 'emotional' (CG = 16%), 'biology and health knowledge' (TG = 9.09%; CG = 16%), and 'ethical knowledge' (CG = 8%) factors.

**Table 1. Framework of analysis applied to assess students' attitudes towards intraspecific diversity.**

| ABC model component | Factors | Description | Examples |
|---|---|---|---|
| Affective | Emotional | The student justifies his/her choice by referring to personal and individual reasons, preferences, experiences, or habits. | ". . . because then I could taste them all (. . .) because I really like beans. . ." (student 5B, pre-test); ". . . because since I was a little girl when I ate feijoada [traditional Portuguese dish] I liked it very much. . ." (student 4A, pre-test); ". . . because it has all the beans I like and some for me to try (. . .) because this way we know what is good for us and what is not. . ." (student 12C, post-test); |
| | Aesthetic | The student justifies his/her choice by referring to preference for aesthetic characteristics, such as color, shape, smell or because it is more beautiful. | ". . . because it is my favorite color, it is black, it is very dark. . ." (student 22C, pre-test); ". . . I think the color is funny. . ." (student 18B, pre-test); ". . . it's more beautiful all varied. . ." (student 7C, post-test); |
| | Social/Cultural | The student justifies his/her choice by referring to the importance of the variety(ies) for his/her relatives or people close to him/her, or for the traditions and customs of the community in which he/she is involved. | ". . . because all my family likes black beans, except me. . ." (student 10A, pre-test); ". . . imagine I have a guest and I only have black beans and the guest doesn't like black beans, so if I have one of each my guest can eat the others, which he/she likes. . ." (student 6A, post-test); |
| Behavioral | Behavioral intention | The student verbally states the bag of seeds he/she would like to show on his/her farm. | "this [student points to the biodiverse bag]. . . because it has more variety. . ." (student 13B, pre-test), "this one [student points to a bag with only one variety]. . . because I really like black beans. . ." (student 6A, pre-test); |
| Cognitive | Biology and health knowledge | The student justifies his/her choice by referring to biology and health topics addressed in the educational activity[1]. | ". . . because if we always eat the same thing we don't know what the others taste like. . ." (student 8B, pre-test); ". . . because when we want a meal we want it with one type of bean, when we want another meal we want it with another type of bean. . ." (student 12B, pre-test); ". . . because there are several types of beans and we can't always eat the same thing in excess, it's bad for us, we should always eat a lot of variety because of the different tastes, (. . .) there could be people who would like some and others who would like others. . ." (student 7C, post-test); |
| | Economic knowledge | The student justifies his/her choice by referring to economic issues related to the opportunity of saving or earning money. | ". . . because if we didn't have them, we would buy them once and have half a bag left over, then another time and we would keep buying them and then we wouldn't spend all [the beans] and this way we save (. . .) if we buy the bag that has the variety this way we save. . ." (student 12C, post-test); ". . . because then I can sell several varieties and people are not always eating the same thing. . ." (student 11A, post-test); |
| | Ethical knowledge | The student justifies his/her choice by referring to the importance of variety for all living beings. He/she recognizes the importance of sowing different varieties to suit different people and animals' preferences, referring to people in general who are not close to him/her or in his/her community. | ". . . because, also people, we are not all the same, because we have variety, there are Chinese people, Japanese people, I think variety is good for everyone. . ." (student 2A, pre-test); ". . . because (. . .) the beans are all varied and so people can choose. . ." (student 13C, pos-test). |

[1] All the biology and health topics explored in the educational activity can be found in S3 Table, as well as examples of the student responses mentioning each of these.

The students who expressed a choice of a bag with only one variety, presented justifications referring to the affective component, predominantly mentioning the 'emotional' factor (TG = 81.82%; CG = 68%), and to a lesser extent the 'aesthetic' (TG = 27.27%; CG = 4%) and the 'social/cultural' (TG = 6.06%; CG = 8%) factors.

## Educational activity's impact on students' attitudes towards intraspecific diversity conservation

When comparing the interview results of the students in the target group, a significant difference in behavioral intention between the pre- and post-test was observed, specifically there

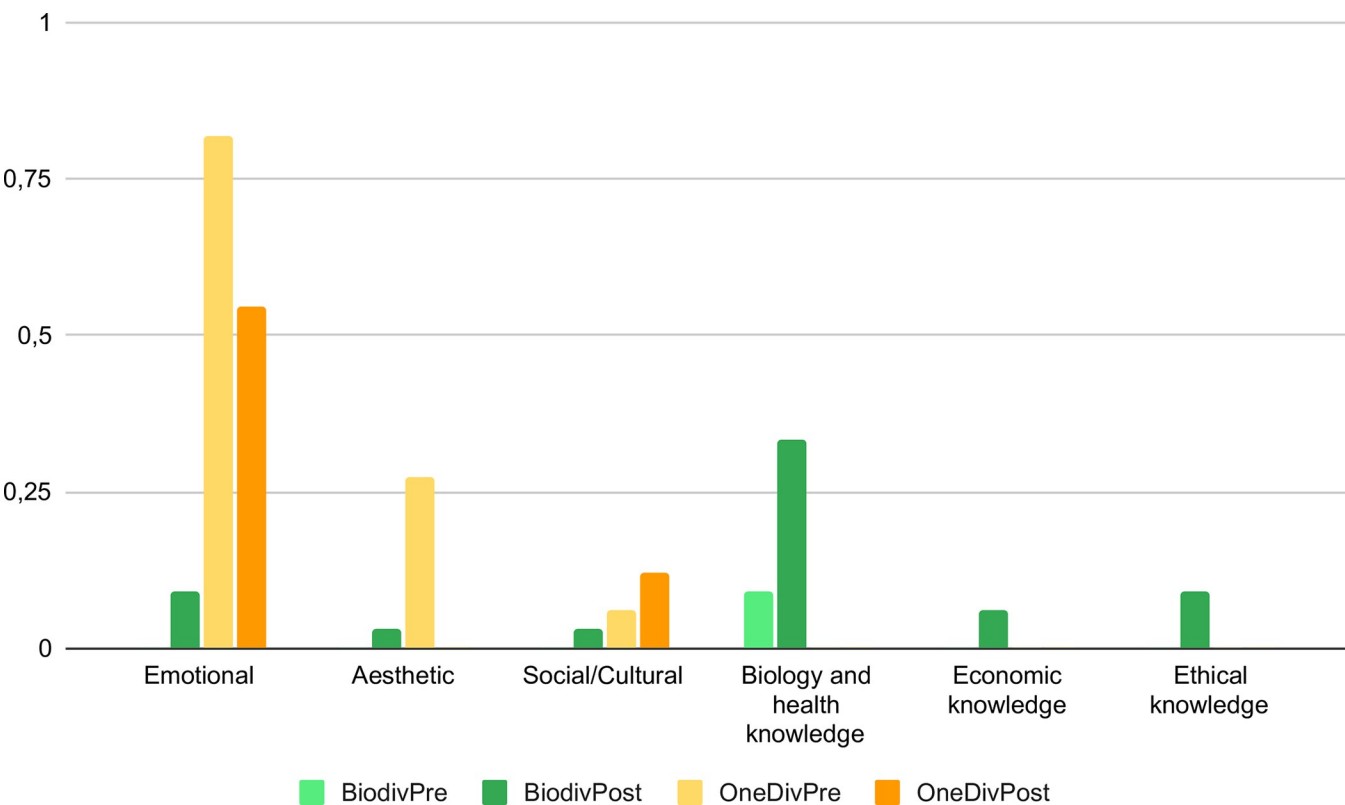

**Fig 3. Frequencies of students' answers assigned to each factor in pre- and post-test according to bag choice.** The light green bars indicate the students of the target group who chose the biodiverse bag in the pre-test (BiodivPre). Dark green bars indicate the students of the target group who chose the biodiverse bag in the post-test (BiodivPost). Light orange bars indicate the students of the target group who chose the bags with one variety in the pre-test (OneDivPre). Dark orange bars indicate the students of the target group who chose the bags with one variety in the post-test (OneDivPost).

was a significant increase in choosing the biodiverse bag ($p$ = .008), with 33.33% of students making this choice in the post-test. No significant differences were observed in the control group (see S1 Table for detailed results). The analysis of the justifications of the students in the target group revealed a significant increase in the mention of the 'biology and health knowledge' factor ($p$ = .008) and a significant decrease in the mention of the 'aesthetic' factor ($p$ = .021) between the pre- and post-test (see Fig 3). The choice of the biodiverse bag in the post-test was followed by justifications that presented mostly justifications referring to the cognitive component of attitude. More specifically, students predominantly mentioned the factor 'biology and health knowledge' (33.33%), but also, to a lesser extent, the factors 'ethical knowledge' (9.09%) and 'economic knowledge' (6.06%). Although less frequently, the factors of the affective component were also mentioned, namely, 'emotional' (9.09%), 'aesthetic' (3.03%), and 'social/cultural' (3.03%).

Among the students in the target group who mentioned the 'biology and health knowledge' factor in the post-test, they specifically mentioned the following topics covered: *i*) different varieties have different properties, such as taste, nutritional properties, or characteristics that make them more suitable for certain dishes (seven students), *ii*) it is healthy to eat distinct varieties of a vegetable and/or it is not healthy to always eat the same variety of a vegetable (six students), and *iii*) different people have distinct tastes and preferences (three students).

The students in the target group who expressed the behavioral intention of choosing a bag with only one variety of beans in the post-test, all mentioned in their justifications factors

related to the affective component of attitudes, namely the 'emotional' (54.55%) and the 'social/cultural' (12.12%) factor.

Chi-square tests showed a significant association of the affective and cognitive factors with the behavioral intention (the bag choice) mentioned by the students of the target group both in the pre and post-tests ($p < 0.01$ for both pre and post-test, see S2 Table for detailed results). In other words, our results showed that the choice of a bag with only one variety tends to be associated with the affective component and the choice of the biodiverse bag tends to be associated with the cognitive component.

When we focus on the change or non-change in behavioral intention from pre- to post-test of the students in the target group, we found that the students who kept the choice of bags with only one variety (OneDivPre-OneDivPost), in the pre-test just mentioned the three factors of the affective component, the post-test showed a decrease in reference to the 'aesthetic' factor (from seven students to zero students) and an increase in reference to the 'social/cultural' factor (from one student to 4 students). The students who kept the choice of the biodiverse bag (BiodivPre-BiodivPost), having mentioned in the pre-test only the 'knowledge approached' factor of the cognitive component, in the post-test they also mentioned the 'ethical knowledge', 'emotional' and 'aesthetic' factors (one student in each factor). The students who changed their choice from a bag with only one variety on the pre-test to the biodiverse bag on the post-test (OneDivPre-BiodivPost), mentioned in the pre-test all the factors of the affective component, and in the post-test decreased the reference to the 'emotional' and 'aesthetic' factors of the affective component, and increased the reference to all the factors of the cognitive component. Namely, the 'emotional' factor from eight students was referred to in the post-test only by two students, the 'aesthetic' factor changed from two students to none, the 'biology and health knowledge' factor from zero students changed to eight students, the 'economic knowledge' factor from none to two students, and the 'ethical knowledge' factor from none changed to two students. None of the students changed their choice from a biodiverse bag to a bag with only one variety. These differences between the factors mentioned according to whether or not the behavioral intention changes can be seen in Fig 4.

## Discussion

With the present study we *i)* developed a valid and reliable framework to evaluate students' attitudes towards intraspecific diversity that is aligned with the literature in the field and adapted to the elementary school students' answers; *ii)* show that, before educational activities, most of the students would choose the non-biodiverse bag, providing arguments mostly related with emotional factors, but also with aesthetic and social/cultural factors to support that choice; and *iii)* support the positive impact of educational activities that explore intraspecific biodiversity of vegetable food species in students' attitudes towards intraspecific diversity.

To the best of the authors' knowledge, this was the first attempt to develop an analysis framework to analyze people's attitudes towards intraspecific diversity conservation. Since this analysis framework was developed with a small number of students from a single elementary school, its implementation in other contexts may reveal the need to include additional categories. Furthermore, this framework has categories of analysis dependent on the learning objectives addressed in the educational activity, namely the knowledge factors of the cognitive component, which gives this framework an adaptive character that will benefit from its implementation in different contexts in the future. Implementation in different contexts may give rise to new categories of analysis and even further refinement of existing categories. This framework may be useful to further explore how students' socio-cultural backgrounds influence their attitudes towards intraspecific biodiversity. It will also be important to further

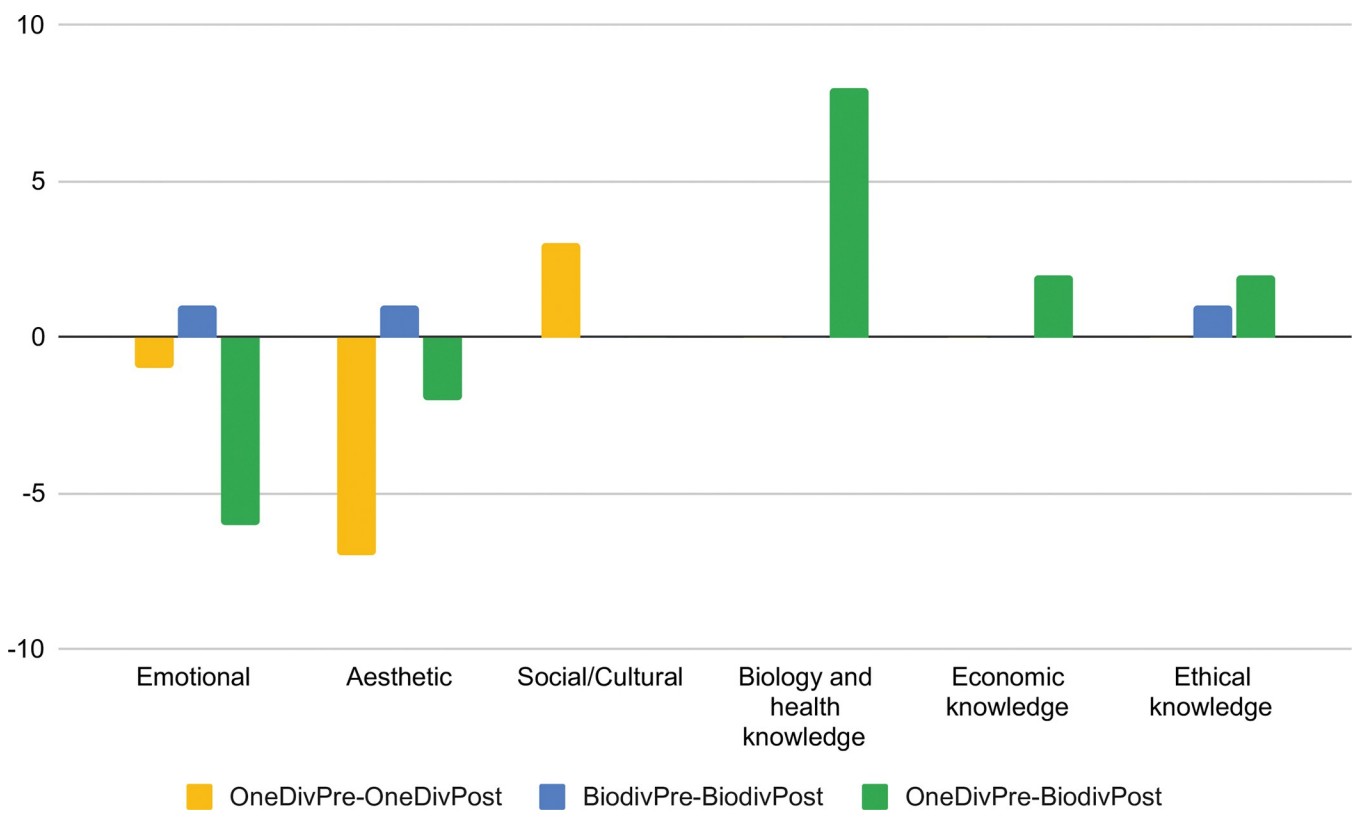

**Fig 4. Difference between the absolute frequencies of students' answers assigned to each factor in the pre- and post-test according to change or non-change in behavioral intention.** Yellow bars indicate the students who chose a bag with only one variety in the pre- and post-test (OneDivPre-OneDivPost). Blue bars indicate the students who chose the biodiverse bag in the pre- and post-test (BiodivPre-BiodivPost). Green bars indicate the students who chose a bag with only one variety in the pre-test and the biodiverse bag in the post-test (OneDivPre-BiodivPost).

evaluate the impact of educational approaches in students' attitudes towards intraspecific biodiversity and to develop more effective educational approaches to enhance public support for biodiversity conservation measures. This is particularly important since intraspecific diversity is essential to ensure food security [16], namely to cope with agricultural losses resulting from global warming, to control pests, and to increase the availability and possibility of intake of various nutrients [21–25].

Regarding students' attitudes towards intraspecific diversity conservation, our results showed that when elementary school students are asked about their behavioral intention, there are few who choose the most biodiverse option. Although our sample is limited to students from a single Portuguese school (and for this reason not generalizable), the fact that this school is located in an urban area provides additional support to the results described by Rosalino et al. [39]. These authors showed that Portuguese students from urban areas are less likely to have a pro-conservation attitude towards biodiversity than students from Brazil or from rural areas. According to our results, the affective and cognitive components of attitudes seem to have different associations with students' behavioral intentions. In fact, the students who chose a bag with only one variety of seeds in the pre-test, referred very often to factors from the affective component, more specifically mostly emotional factors. According to Miralles et al. [65], our emotional connection towards other species has an impact on our attitudes towards them, and this decreases with evolutionary divergence time. But our results further show that a positive attitude towards the conservation of intraspecific diversity is associated

with the cognitive component of attitudes. The students who had the behavioral intention of choosing a biodiverse bag in the pre-test, mostly mentioned biology and health knowledge factors, such as different varieties having different properties and the fact that a varied diet is a healthy habit. Rosalino et al. [39], show that the interests and values related to the students' own health override attitudes towards biodiversity conservation. But in the case of our activity, we associated biodiversity conservation with health knowledge arguments which may have influenced and enhanced the students' choices and positive attitudes towards biodiversity conservation. The use of food species to explore intraspecific diversity may have further facilitated this articulation. Given this, in future studies, it would be interesting to explore whether this type of approach can also have an impact on conservation attitudes towards particular species. A possible example could be the conservation of snakes and other reptiles that usually face strong human persecution [66], but whose venom is used as a source of several medicines [67]. We should however be careful as the traditional use of some species (including snakes) for ethnomedicine was also shown to be detrimental for these species' conservation [68].

Although less frequently, some students also mentioned emotional factors when choosing the biodiverse bag in the pre-test. These results show that students at this grade level can already have a personal preference for intraspecific diversity, and, despite their young age they may have already acquired knowledge about healthy living habits, which can affect their choices. In fact, in Portugal students are aware of the importance of healthy eating from pre-school onwards [43, 69], which may in this case have contributed to the students' acquisition of this prior knowledge. These results are consistent with the results of the questionnaire applied by Schlegel et al. [38] on attitudes towards insects, which concludes that species knowledge is a significant predictor of children's attitudes. Having used foods from the students' daily lives may have influenced their choices due to the high likelihood that they had already acquired some knowledge about the food species used to conduct the interview.

According to our results, exploring the food intraspecific diversity through inquiry based learning and experiential approach seems to promote positive attitudes toward the conservation of intraspecific diversity. After the implementation of the activity, a significant increase in students choosing the biodiverse bag was observed. These positive results towards the conservation of intraspecific diversity after the implementation of an educational activity are similar to other studies' results on children's attitudes towards certain species such as human parasites, birds, reptiles or extensive grassland [34–37]. However, the percentage of students who chose the biodiverse bag in the post-test (33.33%) suggests that the approach could be improved. Future studies may try to explore further the effect of intraspecific diversity on the ability of species to resist diverse ecological conditions (see possible activities at https://bit.ly/3ZgIidD and http://bit.ly/3EwIHkn) and adapt to climate change, a dimension not addressed in this educational activity. In our results, the change in the behavioral intention of the students was frequently associated with the cognitive component factors and mostly the 'biology and health knowledge' factor, reinforcing the impact of our educational activity. Similar results were obtained in previous studies where the learning achieved during educational activities directly and/or indirectly influenced students' attitudes towards specific species [34–37, 70].

A decrease in reference to affective component factors, with a significant decrease in the 'emotional' factor, was also observed. Similar results were obtained by Asshoff et al. [34], in their study on attitudes towards bedbugs. A possible explanation for these results could be the impact of students' involvement on their argumentation skills. In fact, according to the ELM, involvement induces differences in the extent of information processing activity, allowing information to be processed as arguments [33]. Given this, the students' engagement in our educational activity, (designed so that students can assign a personal relevance to the topic, and explore and work with the scientific information), may have influenced their attitude, and

contributed to their ability to present more robust arguments to support their ideas. This may have resulted in the observed increased reference to factors of the cognitive component. On the other hand, by enabling each student to assign personal relevance to the varieties of vegetables that they themselves tasted and classified, the educational activity may have fostered the emotional engagement needed for them to engage more strongly in the processing of arguments favoring the conservation of intraspecific diversity. Given this, it is not possible to say that the affective component is not present in students' attitudes or as these may simply not have been mentioned by them given the acquisition of cognitive arguments that the students may see as stronger or more convincing. To overcome this problem in our study design, in future studies it would be valuable to add questions to the semi-structured interview specifically addressing the emotional component of attitudes, to fully understand the students' justifications. Additionally, our results support that attitudes towards intraspecific diversity may be more easily transferred between species than food preferences [46], since the bean varieties used in the pre- and post-test were never mentioned during the educational activity. The transfer of attitudes towards intraspecific diversity in different species needs to be further studied. Furthermore, it would also be interesting that future studies involve a larger number of participants to test the association between the different components of attitudes towards intraspecific diversity conservation.

In summary, our results support those of other studies [36] that show that active learning activities that requires/fosters the emotional engagement of the students and provides them time for cognitive building around the information that is important for evaluating their attitudes, facilitates attitudinal changes. Our results further suggest that educational activities aimed to promote positive attitudes towards the conservation of intraspecific diversity may attempt to promote the emotional engagement by addressing individual emotional preferences, aesthetic and social/cultural value of biodiversity while providing them time to explore information related with biological/health, economic and ethical knowledge.

Besides the restricted location and the limited number of students, which do not allow the generalization of our findings and highlight the need for additional studies, our study has other limitations that deserve discussion. We used a semi-structured interview to collect the students' answers, however the use of a more structured interview may have benefits for data analysis. Namely, as we previously discussed, adding a question for each component of the attitudes would ensure that all students would express themselves about each of the components, providing a better overview of their attitudes. On the other hand, the replicability of this activity in other populations needs to be studied. The same holds true for the stability of the effect over time of the educational activity's impacts. Our results do not allow us to know how stable is the effect of the activity over longer periods and future studies should consider follow-up studies after longer periods of time.

In future studies, factors such as the frequency of contact with nature or the characteristics of the areas where children live would also be interesting to be analyzed to study if these affect attitudes towards intraspecific diversity conservation, as has been studied for biodiversity conservation [39] or for environmental awareness [71, 72]. Additionally, it would be interesting to study whether students' health and eating habits impact their attitudes towards intraspecific diversity conservation, as this information could contribute to the development of new approaches to biodiversity conservation education. An important additional limitation of our study is related to the lack of information relative to the acceptance and use of activities such as the one presented here by elementary school teachers. Although the activity reported here is aligned with Portuguese official standards for these grade levels it would be important to study if and how elementary school teachers would perform this activity in their classrooms and if the results here reported would hold truth in this case. This would argue for a study on this

educational activity led by teachers, using equipment available in schools, to test its effectiveness in the real world and to enable more teachers to adopt this educational activity.

## Conclusions

Our findings suggest that inviting elementary school students to explore intraspecific diversity of food through an educational activity could be a promising new approach to foster positive attitudes towards intraspecific diversity conservation. This educational activity could be applied to other vegetables or fruits. To the best of the authors' knowledge, this is the first study that has developed an analysis framework to assess students' attitudes towards the conservation of intraspecific diversity, assessed students' attitudes towards the conservation of intraspecific diversity, and evaluated the impact of educational activity on students' attitudes towards the conservation of intraspecific diversity. Despite the limitations to generalization discussed above, this study offers new perspectives that can be further developed and/or refuted by additional studies. Our results revealed that elementary school students when choosing intraspecific diversity mostly justify their choices with reference to factors in the cognitive component of attitudes and when choosing one variety mostly justify their choices with reference to emotional factors in the affective component of attitude. Our educational activity increased the number of students who chose biodiversity and consequently the reference to the knowledge factor, while a decrease in the reference to the emotional factor was observed. This points to new directions for future research on factors that influence children's attitudes toward intraspecific diversity conservation. Moreover, it suggests new approaches to enhancing public support for intraspecific diversity conservation measures in agricultural species that are fundamental to ensure food security.

## Supporting information

**S1 Table. McNemar tests results to test for significant differences between pre- and post-tests results for each of the categories of analysis in control and target groups.** Asterisks (*) denote post-test values that significantly differ from pretests according to McNemar test results (p < .05). K1 to K8 are different biology and health knowledge topics addressed in the educational activity. Namely, K1—Different varieties of a vegetal species have different properties (different varieties have distinct tastes, different varieties may have distinct features that make them more suitable for distinct dishes, different varieties have different nutritional properties and make our diet more diverse); K2—Different varieties may grow and produce differently in distinct environments; K3—The food's degree of ripeness alters its flavor; K4—Different people have distinct tastes and preferences; K5—It is healthy to eat distinct varieties of a vegetable and/or it is not healthy to always eat the same variety of a vegetable; K6—Our tastes change over time, so we should try different varieties of vegetables; K7—The way we chew food influences the taste we get from it; K8—The fact that an individual is sick can change the way they taste food.
(DOCX)

**S2 Table. Chi-square tests results to study possible associations between affective and cognitive components with the students' behavioral intention in pre- and post-tests.** K1 to K8 are different biology and health knowledge topics addressed in the educational activity. Namely, K1—Different varieties of a vegetal species have different properties (different varieties have distinct tastes, different varieties may have distinct features that make them more suitable for distinct dishes, different varieties have different nutritional properties and make our diet more diverse); K2—Different varieties may grow and produce differently in distinct

environments; K3—The food's degree of ripeness alters its flavor; K4—Different people have distinct tastes and preferences; K5—It is healthy to eat distinct varieties of a vegetable and/or it is not healthy to always eat the same variety of a vegetable; K6—Our tastes change over time, so we should try different varieties of vegetables; K7—The way we chew food influences the taste we get from it; K8—The fact that an individual is sick can change the way they taste food. (DOCX)

**S3 Table. Description and examples of the various topics of the 'biology and health knowledge' factor.** Underscore (_) denotes that no examples of the specific topic of the 'biology and health knowledge' factor were found in the students' answers. (DOCX)

## Acknowledgments

We would like to acknowledge the teachers, the students who participated in this study, along with their parents, and the school board team for allowing us to perform this study.

## Author Contributions

**Conceptualization:** Patrícia Pessoa, Sara Aboim, Lisa Afonso, J. Bernardino Lopes, Xana Sá-Pinto.

**Data curation:** Patrícia Pessoa.

**Formal analysis:** Patrícia Pessoa, Sara Aboim, Lisa Afonso, Xana Sá-Pinto.

**Investigation:** Patrícia Pessoa, Sara Aboim, Lisa Afonso, Xana Sá-Pinto.

**Methodology:** Patrícia Pessoa, Sara Aboim, Lisa Afonso, J. Bernardino Lopes, Xana Sá-Pinto.

**Project administration:** Xana Sá-Pinto.

**Supervision:** Xana Sá-Pinto.

**Validation:** J. Bernardino Lopes, Xana Sá-Pinto.

**Visualization:** Patrícia Pessoa.

**Writing – original draft:** Patrícia Pessoa, Sara Aboim, Xana Sá-Pinto.

**Writing – review & editing:** Patrícia Pessoa, Sara Aboim, Lisa Afonso, J. Bernardino Lopes, Xana Sá-Pinto.

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
