## [Decision Letter · Decision Letter 0]

12 Jul 2023

PONE-D-23-11872Tasting to preserve: An educational activity to promote children’s positive attitudes towards intraspecific diversity conservation.PLOS ONE

Dear Dr. Pessoa,

Thank you for submitting your manuscript to PLOS ONE. After careful consideration, we feel that it has merit but does not fully meet PLOS ONE’s publication criteria as it currently stands. Therefore, we invite you to submit a revised version of the manuscript that addresses the points raised during the review process.

We look forward to receiving your revised manuscript.

Kind regards,

Biswajit Pal, M.SC., Ph.D

Academic Editor

PLOS ONE

Journal Requirements:

Additional Editor Comments:

Kindly consider the modifications addressed by the reviewers. Kindly discuss the educational methodology clearly so other researchers can adopt the methodology for future work.

Researchers have done this research in a specific location with a limited number of students. How can you generalize your findings with larger population?

Reviewers' comments:

Reviewer's Responses to Questions

**Comments to the Author**

1. Is the manuscript technically sound, and do the data support the conclusions?

Reviewer #1: Yes

Reviewer #2: Partly

Reviewer #3: Yes

2. Has the statistical analysis been performed appropriately and rigorously? 

Reviewer #1: Yes

Reviewer #2: Yes

Reviewer #3: Yes

3. Have the authors made all data underlying the findings in their manuscript fully available?

Reviewer #1: Yes

Reviewer #2: Yes

Reviewer #3: Yes

4. Is the manuscript presented in an intelligible fashion and written in standard English?

Reviewer #1: No

Reviewer #2: No

Reviewer #3: Yes

5. Review Comments to the Author

Reviewer #1: The manuscript is excellent, in its topic and in the way it was developed. A great, creative idea that was tested based on good methodology. The way it is developed is easy for the reader to follow and any argument was well supported either by their results or by the literature. I found this as a great contribution to the literature.

I only found some typos that need correction as the journal doesn't edit the manuscripts.

Please, check the following typo:

140 “Filling this gap”

153 “specie”

273 “with” to “within”

271-273 You need to mention there which was the unit of analysis

274 I would replace “rated” by “analyzed”

434 Replace “which” with “whose”

469 Replace “students” with “students’ ”

499 Replace “affects” with “affect”

Reviewer #2: The manuscript is a technically sound piece of scientific research as the intraspecific diversity conservation is an important issue nowadays. The data also support conclusion.

Further work must be done with the methodology part. The manuscript does not provide a clear explanation of sampling method and selection criteria.

Data collection procedure and the process of educational activity need to be revised and specific.

The study addresses positive impact of educational activity of elementary school students towards intra specific diversity conservation. It is still unclear whether the activity or framework could be generalized for the other population of the society.

The language of the manuscript is correct and clear. A minor typological error at materials and method part of the manuscript need to be revised.

Reviewer #3: A statistical test may also be added to establish a causal relationship. It may be considered.

However, I have made the required changes to the paper which may also be taken into account. A few grammatical errors have also been marked yellow.

6. PLOS authors have the option to publish the peer review history of their article (what does this mean?). If published, this will include your full peer review and any attached files.

Reviewer #1: No

Reviewer #2: No

Reviewer #3: No

---

## [Author Response · Author response to Decision Letter 0]

15 Sep 2023

We truly appreciate all the comments and suggestions from the editor and reviewers. We believe that all of them have greatly contributed to improve our work. We present below all the changes made to address the concerns and requests we received from you and from the reviewers. We hope we have successfully responded to all of the suggestions and concerns and that our paper can be now considered to be published in PLOS One.

On behalf of the authors’

Patrícia Pessoa

Additional Editor Comments:

Kindly consider the modifications addressed by the reviewers. Kindly discuss the educational methodology clearly so other researchers can adopt the methodology for future work.

Thank you very much for your editorial support and for your suggestion to further discuss the methodology used. To facilitate the future use of this methodology we:

i) provided additional details about the activity in the methodology section (lines 135-136, 154-156, 158-165, 167-169, 170-172, pages 6 and 7 - please see the changes made to these lines described below in response to reviewers' comments/suggestions);

ii) extended the discussion of the paper by adding in lines 480-487 the following text “In summary, our results support those of other studies (36) that show that active learning activities that requires/fosters the emotional engagement of the students and provides them time for cognitive building around the information that is important for evaluating their attitudes, facilitates attitudinal changes. Our results further suggest that educational activities aimed to promote positive attitudes towards the conservation of intraspecific diversity may attempt to promote the emotional engagement by addressing individual emotional preferences, aesthetic and social/cultural value of biodiversity while providing them time to explore information related with biological/health, economic and ethical knowledge.”

Researchers have done this research in a specific location with a limited number of students. How can you generalize your findings with larger population?

We truly appreciate the editor's question. This is a clear limitation of our study and to further emphasize this limitation we have added:

- In lines 384-386 the following text “Since this analysis framework was developed with a small number of students from a single elementary school, its implementation in other contexts may reveal the need to include additional categories.” 

- In lines 402-405 the following text “Although our sample is limited to students from a single Portuguese school (and for this reason not generalizable), the fact that this school is located in an urban area provides additional support to the results described by Rosalino et al. (39).” 

- In lines 488-490 the following text “Besides the restricted location and the limited number of students, which do not allow the generalization of our findings and highlight the need for additional studies, our study has other limitations that deserve discussion.”

- In lines 522-524 the following text “Despite the limitations to generalization discussed above, this study offers new perspectives that can be further developed and/or refuted by additional studies.”

Through these changes, we have attempted to clarify that, since our study had a specific location and a limited number of students, our results - both the framework developed and the impact of the educational activity - cannot be generalized for the whole population of society and that additional studies with a higher number of students and coming from more diverse cultural backgrounds are needed. 

Reviewers' comments:

Comments to the Author

Reviewer #1: The manuscript is excellent, in its topic and in the way it was developed. A great, creative idea that was tested based on good methodology. The way it is developed is easy for the reader to follow and any argument was well supported either by their results or by the literature. I found this as a great contribution to the literature.

 I only found some typos that need correction as the journal doesn't edit the manuscripts.

 Please, check the following typo:

 140 “Filling this gap”

 153 “specie”

 273 “with” to “within”

 271-273 You need to mention there which was the unit of analysis

 274 I would replace “rated” by “analyzed”

 434 Replace “which” with “whose”

 469 Replace “students” with “students’ ”

 499 Replace “affects” with “affect”

All typos were corrected according to the suggestions of Reviewer #1. In order to address the request on lines 256-259, the following text has been added: “The unit of analysis was the “meaning unit”, which is defined as “the constellation of words or statements that relate to the same central meaning” (Graneheim and Lundman 2004, p. 106). In our case, a meaning unit can consist of a sentence or sentence segment that expresses an idea and aligns with a specific category of the framework of analysis.”

Reviewer #2: The manuscript is a technically sound piece of scientific research as the intraspecific diversity conservation is an important issue nowadays. The data also support conclusion.

 Further work must be done with the methodology part. The manuscript does not provide a clear explanation of sampling method and selection criteria.

In order to clarify our methodology and address the reviewer's request, we have added: 

- In lines 122-125 the following text “We invited third-grade children (8-13-year-old) from one public elementary school in the northern region of Portugal, chosen by convenience (Cohen et al., 2018), to attend the educational activity and answer the short interview described below (Girls: 41.6%; Age: M = 8.88; SD = 0.65).” 

- In lines 128-130 the following text “The children were randomly distributed (1:1) within the classrooms, using a random number generator, to form a target group (n=33) and a control group (n=25).”

Data collection procedure and the process of educational activity need to be revised and specific.

To specify our data collection procedure, the following text has been added to the lines 216-221: “After the first verbalization of the choice and justification, the researcher asked follow-up questions and probing questions (Cohen et al., 2018) to encourage the students to elaborate more on their justification and, whenever possible, to explain their choice in more depth. These questions were always supplemented by interpretation questions (Cohen et al., 2018) to ensure that the researcher understood the student's justification. Interviews ended with a question to confirm that there were no reasons not previously mentioned to justify the student's choice.”

Revisions were made to the description of the educational activity. In particular, the following has been added: 

- Lines 135-136 - “The educational activity was designed for third-grade students following an inquiry based learning and experiential approach (Pedaste et al., 2015)”.

- Lines 154-156 - “In the second session, students analyzed the bar graphs and postulated hypotheses to explain two observations that were posed to them to be answered through an inquiry-based learning approach (Pedaste et al., 2015) …”

- Lines 158-165 - “In groups of 3 to 5, students were invited to plan an experiment to test their chosen hypothesis (examples of hypotheses that were tested can be found in (46)) using a worksheet adapted from an official Portuguese educational set (Martins, 2006). The worksheet asked them to describe the hypothesis rephrased as a question, what would need to be kept constant to test the hypothesis, what would vary, what was going to be measured, how were they going to register data, what would be the experimental procedure, what was needed to perform such experimental procedure, what were the predicted and observed results, and what conclusions could be taken.”

- Lines 167-169 - “The devices used to measure sugar (refractometer) and pH (pH sensor) were shown to the students whenever necessary (Martins, 2006).”

- Lines 171-172 - “(examples of experimental procedures used to test each hypothesis can be found in (46)).”

The study addresses positive impact of educational activity of elementary school students towards intra specific diversity conservation. It is still unclear whether the activity or framework could be generalized for the other population of the society.

We attempted to respond to this comment through our previous response to the editor's comment. Please see the second reply to the editor's comments above. 

The language of the manuscript is correct and clear. A minor typological error at materials and method part of the manuscript need to be revised.

All the typos identified in the attached file have been corrected. 

Reviewer #3: A statistical test may also be added to establish a causal relationship. It may be considered.

Thank you for this suggestion. In fact, we considered including a statistical test to establish a causal relationship, but given the limited number of participants in the study this was not possible, since the number of evidence presented for each category is not large enough to carry out more statistical tests than those already used and described. 

However, I have made the required changes to the paper which may also be taken into account. A few grammatical errors have also been marked yellow.

All the typos identified in the attached file have been corrected. 

Additional notes: 

1) In order to address the suggestions and comments of the editor and reviewers, three references were added to our manuscript:

Pedaste M, Mäeots M, Siiman LA, De Jong T, Van Riesen SAN, Kamp ET, et al. Phases of inquiry-based learning: Definitions and the inquiry cycle. Educ Res Rev. 2015 Feb;14:47–61.

Martins IP. Exploring objects: Fluctuation in fluids. Didactic guide for teachers. [Internet]. Available from: https://www.dge.mec.pt/sites/default/files/Basico/Documentos/ explorando_flutuacao_liquidos.pdf

Graneheim UH, Lundman B. Qualitative content analysis in nursing research: concepts, procedures and measures to achieve trustworthiness. Nurse Educ Today. 2004 Feb;24(2):105–12.

2) The access numbers or DOI required to access the repository of our data will be provided if the manuscript is accepted for publication.

---

## [Decision Letter · Decision Letter 1]

17 Oct 2023

Tasting to preserve: An educational activity to promote children’s positive attitudes towards intraspecific diversity conservation.

PONE-D-23-11872R1

Dear Dr. Pessoa,

We’re pleased to inform you that your manuscript has been judged scientifically suitable for publication and will be formally accepted for publication once it meets all outstanding technical requirements.

Kind regards,

Biswajit Pal, M.SC., Ph.D

Academic Editor

PLOS ONE

---

## [Editor Report · Acceptance letter]

24 Nov 2023

PONE-D-23-11872R1 

Tasting to preserve: An educational activity to promote children’s positive attitudes towards intraspecific diversity conservation. 

Dear Dr. Pessoa:

I'm pleased to inform you that your manuscript has been deemed suitable for publication in PLOS ONE. Congratulations! Your manuscript is now with our production department. 

Kind regards, 

on behalf of

Dr. Biswajit Pal 

Academic Editor

PLOS ONE